# Epigenetic Regulation by microRNAs in Hyperhomocysteinemia-Accelerated Atherosclerosis

**DOI:** 10.3390/ijms232012452

**Published:** 2022-10-18

**Authors:** Raquel Griñán, Joan Carles Escolà-Gil, Josep Julve, Sonia Benítez, Noemí Rotllan

**Affiliations:** 1Institut d’Investigació Biomèdica Sant Pau (IIB SANT PAU), 08041 Barcelona, Spain; 2Departament de Bioquímica i Biologia Molecular, Universitat Autònoma De Barcelona, 08193 Barcelona, Spain; 3CIBER de Diabetes y Enfermedades Metabólicas Asociadas (CIBERDEM), Instituto de Salud Carlos III, 28029 Madrid, Spain

**Keywords:** hyperhomocysteinemia, microRNAs, endothelial cells, macrophages, vascular smooth muscle cells and atherosclerosis

## Abstract

Increased serum levels of homocysteine (Hcy) is a risk factor for cardiovascular disease and is specifically linked to various diseases of the vasculature such as atherosclerosis. However, the precise mechanisms by which Hcy contributes to this condition remain elusive. During the development of atherosclerosis, epigenetic modifications influence gene expression. As such, epigenetic modifications are an adaptive response to endogenous and exogenous factors that lead to altered gene expression by methylation and acetylation reactions of different substrates and the action of noncoding RNA including microRNAs (miRNAs). Epigenetic remodeling modulates cell biology in both physiological and physiopathological conditions. DNA and histone modification have been identified to have a crucial role in the progression of atherosclerosis. However, the potential role of miRNAs in hyperHcy (HHcy)-related atherosclerosis disease remains poorly explored and might be essential as well. There is no review available yet summarizing the contribution of miRNAs to hyperhomocystein-mediated atherogenicity or their potential as therapeutic targets even though their important role has been described in numerous studies. Specifically, downregulation of miR-143 or miR-125b has been shown to regulate VSCMs proliferation in vitro. In preclinical studies, downregulation of miR-92 or miR195-3p has been shown to increase the accumulation of cholesterol in foam cells and increase macrophage inflammation and atherosclerotic plaque formation, respectively. Another preclinical study found that there is a reciprocal regulation between miR-148a/152 and DNMT1 in Hcy-accelerated atherosclerosis. Interestingly, a couple of studies have shown that miR-143 or miR-217 may be used as potential biomarkers in patients with HHcy that may develop atherosclerosis. Moreover, the current review will also update current knowledge on miRNA-based therapies, their challenges, and approaches to deal with Hcy-induced atherosclerosis.

## 1. Introduction

In clinical practice, hyperhomocysteinemia (HHcy) is generally defined as fasting circulating homocysteine (Hcy) concentrations >15μM. HHcy is a medical condition of elevated homocysteine (Hcy), a non-protein-coding amino acid, which is prevalent in the general population, and it has severe implications for human health. Mild-moderate elevations of Hcy are not uncommon, occurring in 5–7% of individuals. Cardiovascular disease (CVD) and stroke [1], dementia [2], schizophrenia [3], infertility and birth defects [4], cancer [5,6], liver injury [7], and osteoporosis [8] are all associated with HHcy. Particularly, plasma elevations of Hcy is considered an independent risk factor for atherosclerosis and venous thrombosis [9]. Plasma Hcy concentrations are strongly influenced by various non-genetic causes as well as genetic determinants. Interestingly, mechanistic studies have shown that Hcy can exert a variety of effects expected to promote CVD and thrombosis, including increased expression of adhesion molecules, tissue factors, cytokines, blood coagulation factor V, inhibition of fibrinolysis, increased platelet reactivity, and disruption of nitric oxide metabolism. Other factors that have been identified to have a crucial role in the progression of HHcy-related atherosclerosis are epigenetic regulation via DNA methylation or histone modification. However, the potential role of miRNAs, another class of epigenetic regulators, in HHcy-related atherosclerosis disease remains poorly explored and might be critical as well. Thousands of miRNAs have been described since their discovery in 1998 in both healthy and pathological states, being involved in nearly every biological pathway. It is known that they are regulating and controlling gene expression in most organisms [10,11] and have the ability to target multiple genes, generally within the same or related pathway. miRNAs function as post-transcriptional regulators providing a ubiquitous mechanism for control of gene expression either via cleavage and degradation of target mRNA or by inhibition of the translation process [12]. Of note, miRNA therapies exist and may be a better option for Hcy treatment since vitamin B supplementation has partially failed [13,14,15]. Thus, the goal of this review was to offer an updated state-of-the-art study of the miRNAs as epigenetic regulators for these diseases and their possible use as a therapeutic tool.

### 1.1. Homocysteine Biosynthesis and Catabolism

Hcy is a non-essential, sulfur-containing amino acid that is produced in all cells during the methionine cycle. Although it is not directly involved in protein synthesis, Hcy serves as a key intermediate metabolite in methylation reactions [16]. 

The only pathway for Hcy biosynthesis in humans is the transmethylation of the essential, diet-derived amino acid methionine (Met). The transmethylation of Met includes three sequential steps. First, Met is ATP-activated and transformed into S-adenosylmethionine (SAM or AdoMet) through a reaction catalyzed by SAM synthetase (also known as methionine adenosyltransferase (MAT)). Second, SAM is converted into S-adenosylhomocysteine (SAH or AdoHcy) via a specific methyltransferase (MT)-mediated reaction [17]. SAM serves as a universal methyl donor to a variety of acceptors, such as DNA, RNA, proteins, and phospholipids, thus representing a crucial step in the biosynthesis of numerous biochemical compounds [18]. Moreover, SAM has been shown to play a key role in epigenetic mechanisms (that play an important role in gene expression and regulation), including DNA methylations and histone modifications [19]. Third, SAH is rapidly hydrolyzed into Hcy and adenosine by the action of SAH hydrolase. Interestingly, this hydrolysis is a reversible reaction that favors the formation of SAH. The SAM/SAH ratio defines the methylation potential of a cell. Under physiological conditions, Hcy and adenosine need to be rapidly metabolized or exported out of the cell to prevent its accumulation [20]. In case Hcy accumulates, SAH will accumulate as well, acting as a potent inhibitor of MT enzymes and consequently exhibiting an inhibitory effect on the transmethylation reaction of Met [21]. 

Metabolism of Hcy is at the intersection of two alternative pathways: remethylation and transsulfuration. In the remethylation pathway, Hcy acquires a methyl group from either N-5-methyltetrahydrofolate (5-methylTHF) or betaine to form Met. In the first mechanism, 5-methylTHF is the substrate to methylate Hcy, employing cobalamin (Cbl, also known as vitamin B_12_) as the cofactor. The cobalamin-dependent enzyme methionine synthase (MS) catalyzes the methyl transfer from 5-methylTHF to Hcy, forming Met and tetrahydrofolate (THF). At this point, Hcy metabolism is biochemically linked to the intracellular folate cycle. The resulting THF is further converted into 5,10-methyleneTHF by the action of serine hydroxymethyltransferase (SHMT) and ultimately reduced into 5-methylTHF by the enzyme 5,10-methylenetetrahydrofolate reductase (MTHFR) [22,23]. Thus, both folate and vitamin B_12_ play an important role in the regulation of Hcy balance within the cell. In the second mechanism, Hcy is recycled to Met by the enzyme betaine-homocysteine S-methyltransferase (BHMT), using betaine as a cofactor. Whereas the folate-dependent remethylation pathway occurs in all tissues, this betaine-dependent remethylation mainly occurs in the liver and kidney [24].

Alternatively, Hcy can be irreversibly catabolized to cysteine through the transsulfuration pathway. Transsulfuration is sequentially catalyzed by the enzymes cystathionine β-synthase (CBS) and cystathionine γ-lyase (CSE), which are dependent on pyridoxal-5′-phosphate (derived from vitamin B_6_). CBS allows the condensation of Hcy with serine, and the resulting cystathionine is converted into cysteine and α-ketobutyrate. In addition to protein synthesis, cysteine is used in the synthesis of glutathione, an important endogenous antioxidant and an essential compound in the detoxification of many xenobiotics. Excess cysteine is oxidized to inorganic sulfates or taurine, and a major part is excreted in the urine [25]. Schematic representation for Hcy metabolism (Figure 1). 

### 1.2. Hyperhomocysteinemia as a Risk Factor in CVD

HHcy is defined as a medical condition characterized by an abnormally high level of Hcy in plasma. Under physiological conditions, the total concentration of Hcy in plasma of healthy human subjects ranges from 5 to 15 µM, with this baseline value being maintained via constant clearance by the kidney [26]. HHcy occurs as the result of an imbalance between the production and disposal of Hcy. It is generally classified into three levels: mild/moderate (15–30 µM), intermediate (30–100 µM), and severe (greater than 100 µM) [27]. 

A large number of epidemiological studies suggest that HHcy is an independent risk factor for cardiovascular diseases (CVD), including atherosclerosis [28], hypertension [29], vascular calcification [30], and aneurysm [31]. The observation that plasma Hcy was a potential cause of premature vascular disease was first suggested by Dr. Kilmer McCully in 1969. The study was based on pathological findings in infants with HHcy resulting from inborn metabolism deficiency [32]. These findings provoked extensive research that supported the association of HHcy with various vascular pathologies and prompted the study of elevated Hcy-induced vascular effects. 

There are several causes of HHcy that can be categorized into five groups: mutations in genes encoding enzymes that are implicated in Hcy metabolism, nutritional deficiencies, excessive intake of Met, and certain diseases, lifestyle, and the ingestion of certain drugs [33]. Genetic errors of enzymes involved in Hcy metabolism are rare but cause severe forms of HHcy. Particularly, the most common enzyme defect associated with intermediate HHcy is a point mutation in the coding region of the gene for MTHFR [34]. Further, classic homocystinuria due to homozygous deficiency of CBS accounts for the most common severe form of HHcy, and it has been associated with an increased risk of arterial and venous thromboembolic events [35,36]. These patients represent a very small proportion of the general population (1/200,000) and can reach plasma Hcy levels up to 400 µM. Other rarer causes of severe HHcy include homozygous deficiency of MTHFR, deficiency of MS, and impaired activity of MS due to genetic disorders of vitamin B_12_ metabolism [37]. 

In contrast, mild-to-moderate HHcy occurs more frequently (5–7% of the population) and is typically caused by a nutritional deficiency of vitamin B_12_, vitamin B_6_, and folate. Several studies have shown a correlation between low levels of vitamin B_12_ in plasma and a high risk to develop type 2 diabetes, dyslipidemia, and CVD [26]. Folate deficiency has been associated with cardiovascular diseases, cognitive impairment, and cancer [38]. Several diseases such as end-stage renal disease, severe hepatic dysfunction, hypothyroidism, pernicious anemia, diabetes mellitus, and cancer as well as various drugs, alcohol, tobacco, older age, and menopause are believed to be associated with moderate HHcy [39].

### 1.3. Atherosclerosis-Related Hyperhomocysteinemia

Atherosclerosis is a complex disease characterized by the accumulation of lipids and fibrous elements in the large arteries. It is considered a chronic inflammatory disease that results from interactions between modified lipoproteins and cells of the arterial wall, including endothelial cells (ECs), monocytes/macrophages, and vascular smooth muscle cells (VSMCs) [40]. Although there is a clear relationship between Hcy and atherosclerosis [41,42,43,44], the underlying mechanisms that mediate the adverse effects of HHcy are only partly understood. To date, various explanations have been proposed, including HHcy-induced oxidative stress, inflammation, endoplasmic reticulum (ER) stress, and protein N-homocysteinylation, altogether leading to endothelium dysfunction and activation of smooth muscle cell and macrophages [45]. 

It is known that ECs maintain vascular wall homeostasis by regulating vascular tone, permeability, inflammation, and cell growth. Therefore, damage to the endothelium can trigger crucial consequences for vascular stability and function, inducing a cascade of inflammatory events and the subsequent formation of atherosclerotic lesions [46]. In vitro and in vivo studies show evidence that Hcy can impair the ability of ECs to regulate vascular tone [47]. Specifically, Hcy interferes with the production of nitric oxide (NO), a potent endogenous vasodilator [48,49,50]. Decreased bioavailability of NO contributes to thrombosis and atherosclerosis through increased platelet and endothelial activation [51,52,53]. Furthermore, Hcy may deregulate the hydrogen sulfide (H_2_S) signaling pathway, another essential endothelial gasotransmitter [54]. Both NO and H_2_S pathways are closely connected in maintaining vascular homeostasis [55]. 

Additionally, endothelial dysfunction can be evoked by HHcy-generated reactive oxygen species (ROS) yielding oxidative stress in both circulating leukocytes and vascular resident cells. Oxidative stress may result from an accumulation of ROS, including superoxide anion (O_2_^−^), hydrogen peroxide (H_2_O_2_), and peroxynitrite (ONOO^−^) [56]. There have been proposed different molecular mechanisms that mediate HHcy-induced cellular oxidative stress either by activating oxidant systems or by impairing the antioxidant capacity of the cell [57]. One of the mechanisms proposed is Hcy autoxidation in circulation, caused by its highly reactive thiol group, in the presence of transition metals (Fe and Cu), to form homocysteine and H_2_O_2_. However, the role of the autoxidation of Hcy to induce endothelial damage has been questioned since other thiols such as cysteine, whose concentration in plasma is much higher than that of Hcy, can also undergo autoxidation but are not considered a risk factor for cardiovascular diseases [58]. In contrast, most studies show that HHcy-induced cellular oxidative stress is likely to be partly caused by the upregulation of nicotinamide adenine dinucleotide phosphate (NADPH) oxidase (NOX), which is the main source of O_2_^−^ and H_2_O_2_ [59]. Specifically, HHcy was shown to upregulate the expression of NOX in an in vivo mouse model [60] and significantly increased the expression of NOX2 and NOX4 in human ECs [61].

High Hcy levels have been associated with the activation of calcium signaling and increased oxidation in mitochondria, resulting in impaired redox balance and accumulation of O_2_^−^ [62,63]. Moreover, O_2_^−^ can be rapidly converted into H_2_O_2_ via superoxide dismutase (SOD) and further induces the production of other oxygen-free radicals such as ONOO^−^ by reacting with NO, thereby reducing the bioavailability and activity of NO. Likewise, Hcy can decrease NO production by inducing endothelial nitric oxide synthase (eNOS) uncoupling and upregulating inducible nitric oxide synthase (iNOS) in ECs [64,65]. Decreased expression and activity of antioxidant enzymes is indirectly involved in HHcy-induced vascular injury. In particular, the damage to the antioxidant system is linked to the high state of oxidative stress. HHcy-mediated oxidative stress has been associated with the loss of the activity of glutathione peroxidase (GPx), SOD, catalase, thioredoxin, and heme oxygenase-1 (HO-1), decreasing the ability of cells to scavenge the ROS and promoting its accumulation [66,67,68]. Consequently, HHcy-induced oxidative stress can disturb lipoprotein metabolism, promoting low-density lipoprotein (LDL) oxidation and growth of atherosclerotic vascular lesions. Moreover, an increase in oxidative stress can stimulate the expression of cytokines and adhesion molecules in endothelial cells via the NF-κB pathway, which results in pro-inflammatory gene expression and vascular inflammation. The subsequent release of inflammatory cytokines by ECs includes interleukin-6, interleukin-8, and tumor necrosis factor-α [56]. In addition, some studies have revealed that Hcy induces apoptosis in endothelial cells [69], which is a hallmark of atherosclerotic lesions and contributes to the formation and rupture of atherosclerotic plaque. 

Another mechanism by which Hcy induces vascular inflammation could be through the induction of endoplasmic reticulum (ER) stress, as an increase in ER oxidoreductin-1α (Ero1α) expression was observed in the thoracic aorta of HHcy mice in association with ER stress and endothelial inflammation [47]. ER stress can induce disordered lipid metabolism, inflammation, and apoptosis, which are all fundamental processes that contribute to the development and progression of atherosclerosis [70]. 

Likewise, disruption of Hcy metabolism may impact the endothelium through irreversible protein N-homocysteinylation. During protein biosynthesis, Hcy can undergo erroneous cyclization to form Hcy thiolactone (HTL), which reacts with proteins by forming amide bonds with amino groups of lysine residues. Protein N-homocysteinylation has been shown to cause alteration of the biological activity of proteins. Specifically, HTL can interact with LDL, causing aggregation and increase in density and facilitating the uptake by macrophages to form foam cells [71]. Interestingly, one study showed that the compositional changes in Hcy-LDL from diabetic subjects have cytotoxic effects on human endothelial cells, and these alterations of the endothelium and oxidative damage might be the reason of the pathogenesis of diabetic vascular complications [72]. In fact, other studies have observed a relationship between Hcy levels and chronic complications of diabetes, and it has been shown that Hhcy is associated with CVD in diabetes [73,74]. Moreover, epidemiological studies have linked HHcy to type 2 diabetes (T2D) and insulin resistance. It has been shown that Hcy disrupts insulin signaling pathway by decreasing tyrosine phosphorylation of insulin receptors and IRS-1 while increasing serine phosphorylation of IRS-1; and Akt phosphorylation and PI3K activities are also inhibited by Hcy both in vitro and in vivo [75,76,77]. Interestingly, one study demonstrated that polyunsaturated fatty acid (PUFA) supplementation decreases plasma Hcy in diabetic dyslipidemia treated with a statin–fibrate combination [78], and more recently, another study has shown that PUFA has regulatory effects on mRNA expression of key genes involved in Hcy metabolism [79]; however, more studies are needed to investigate the potential mechanism by which n-3 PUFA decreases plasma Hcy concentration. Lastly, HHcy can significantly stimulate the proliferation of VSMCs and the irregular formation of the extracellular matrix, which are prominent features of atherosclerosis. Under physiological conditions, VSMCs are contractile and non-proliferative, allowing for the elasticity and structural integrity of the medial layer. However, elevated blood levels of Hcy promote the expression of adhesion molecules, chemokine, and the VSMC mitogen, leading to a vascular remodeling process that involves VSMCs proliferation, activation of matrix metalloproteinases (MMPs), and induction of collagen synthesis [80]. 

## 2. Epigenetic Regulation of Hyperhomocysteinemia-Related Atherosclerosis 

As previously described, HHcy has been regarded as an emerging risk factor for the development of atherosclerosis [81,82]. Apart from processes such as inflammation, lipoprotein oxidation, and immunity, epigenetic modifications have been reported to regulate some of the genes implicated in extracellular matrix formation, apoptosis, and inflammation [83]. Thus, epigenetic alterations are involved in plaque formation and progression and atherosclerosis-related disease. Many experiments in vitro have shown that Hcy increases VSMC proliferation and induces oxidative damage, which leads to a pro-thrombotic state. Moreover, high levels of Hcy in plasma represent an independent risk factor for premature atherosclerosis, as explained before. However, the mechanism of Hcy in the development of atherosclerosis has not been clarified, and recent data support that the pathogenic role of HHcy in atherosclerosis is due to epigenetic alterations. Epigenetics refers to heritable traits that are not a consequence of DNA sequence. There are three main classes of epigenetic regulation: DNA methylation, histone modification, and noncoding RNA action. The importance of microRNA (miRNAs), which is one of the most studied noncoding RNA, in the HHcy-atherosclerosis field remains poorly explored. Thus, in this review, we focus on the role of miRNAs in HHcy-atherosclerosis-related disease and, in some cases, on their interactions with DNA methylation and histone modifications as well since miRNAs can also be epigenetically regulated [84]. 

### miRNA Regulation of Hyperhomocysteinemia-Related Atherosclerosis

miRNAs are small, single-stranded, endogenous, noncoding RNA containing 18–22 nucleotides that play central roles in a broad range of biological processes [12,85]. miRNAs function as post-transcriptional regulators providing a ubiquitous mechanism for control of gene expression either via cleavage and degradation of target mRNA or by inhibition of the translation process. miRNA function mostly through canonical base pairing between the seed sequence of the miRNA and its complementary seed match sequence, present in the 3′ untranslated regions (3′UTR). miRNAs are the center of attention in molecular and cell biology research. Aberrant expression of miRNAs has been implicated in the pathophysiological processes underlying the development of atherosclerosis and CVD, including changes in endothelial function, vascular smooth muscle cell proliferation and migration, macrophage function, and foam cell formation [10,11].

Several studies indicate that the pathogenic role of HHcy in vascular disease might be triggered by epigenetic alterations of miRNAs. This potential involvement of miRNAs in preclinical models of HHcy-mediated atherosclerosis is summarized in Table 1.

Zhang et al. demonstrated for the first time that the upregulation of DNMT3a directly results in the hypermethylation of miR-143 genes in Hcy-induced VSMCs and a downregulation of its function, resulting in a proliferation of VSCMs, are known to be one of the key processes in the pathogenesis and progression of atherosclerosis [86]. However, it would be interesting to perform a preclinical atherosclerosis study to corroborate the existence of this regulatory circuit involving miR-143 and DNMT3 in vivo. The same laboratory also described another possible signal pathway that regulates VSCMs proliferation induced by Hcy, which is the miR-125b-DNMT3b-p53, but again, more preclinical studies are needed to really prove its role during atherosclerosis [87]. The modification of histones by methylation of the lysine residues has been shown to regulate gene expression by affecting chromatin structure. Xiaoling et al. demonstrated elevated H3K27me3 modification in aortas with Hcy-induced atherosclerotic plaques, and this was accompanied by alterations in the corresponding histone methyltransferase EZH2, which was regulated by miR-92a [88]. However, further investigation is needed for the potential diagnostic or therapeutic target of the miR-92a and EZH2. Interestingly, Yang et al. deciphered the interaction between *DNMT1*, a DNA methyltransferase, and miR-148a/152 in the context of HHcy-related atherosclerosis [89]. Specifically, the authors found that *DNTM1* overexpression results in the hypermethylation of miR-148a and miR-152 genes, leading to the downregulation of miR-148a and miR-152 levels in foam cells treated with Hcy. The authors also observed that miR-148a/152 reversely regulates *DNMT1* gene expression in the context of Hcy-accelerated atherosclerosis, and in vitro transfection of miR-148a/152 promoted lipid accumulation by silencing *DNMT1* in foam cells. Thus, this interesting study defined the reciprocal regulation between miR-148a/152 and *DNMT1* in Hcy-accelerated foam cell differentiation and atherosclerotic lesion [89].

Recent findings demonstrated that Hcy-mediated miR-195-3p promotes macrophage inflammation and the progression of atherosclerosis by targeting IL-31. The reasons for its downregulation are DNA methylation and H3K9 deacetylation catalyzed by DNTM3a and HDAC11, respectively. Specifically, the authors demonstrate that Sp1 interacts with DNMT3a, which allows HDAC11 to bind to miR-195-3p promoter [90]. Restore of miR-195-3p expression might be beneficial for HHcy-related atherosclerosis patients as the authors suggested, but further studies are needed.

On the other hand, it is relevant to mention that miRNAs might be used as biomarkers of atheroma plaque formation and progression in patients with high levels of Hcy. Results from Liu et al. indicated that the levels of atherosclerosis-associated circulating miR-143 and miR-145 are linked to HHcy and suggested that miR-143 may be used as a potential biomarker of HHcy although the number of patients of this study was small [91]. In a recent clinical study, the same laboratory evaluated for the first time atherosclerosis-specific circulating miRNAs expression profiles in patients with HHcy [92]. Specifically, they found that miR-217 may be helpful in predicting the progress of atherosclerosis in HHcy patients [92]. Again, more research and prospective large-scale studies are needed in order to use miRNAs as possible biomarkers in patients with high levels of Hcy that may develop atherosclerosis.

## 3. Target Therapies in HHcy-Related Atherosclerosis

Several randomized trials have evaluated the impact of decreasing serum Hcy among patients with CVD. Remarkably, several secondary prevention trials have shown that B vitamin replacement therapy, which effectively lowers plasma Hcy concentration in patients with moderate HHcy, does not lower cardiovascular risk [13,14,15]. However, there is a great deal of controversy with some of the results from those trials. Some potential causes responsible for the lack of consensus could be differences between study groups with respect to baseline Hcy concentrations, inclusion of patients from countries that do not have regulations regarding food enrichment with folic acid, trial period, patient medication, composition of vitamin formulations, and gender distribution. Importantly, based on recent findings and new analysis of these previous studies, B vitamins or folate supplementation effectively lowers Hcy sasserum levels, and they do prevent stroke in primary prevention studies [93,94,95]. It is possible that this was obscure in early trials by patients with impaired renal function. There is a controversy with the American Heart Association guideline, according to some authors, about the evidence that B vitamins should be used to prevent stroke, which form of folate is the more appropriate, and the use of methylcobalamin or hydroxycobalamin instead of cyanocobalamin [96,97]. Another reason why vitamin B therapy did not work as expected is that maybe the pathological changes already induced by Hcy would be unaffected by thes different plasma Hcy-lowering treatments. Thus, the regulation of those miRNAs that have been shown to be important in HHcy-related atherosclerosis might be a relevant target therapy.

As we reviewed, several miRNAs play a key role during HHcy-related atherosclerosis disease (see Table 1). Mechanistic studies have shown that homocysteine can exert a variety of effects expected to promote CVD and thrombosis, including increased expression of adhesion molecules, tissue factors, cytokines, blood coagulation factor V, inhibition of fibrinolysis, increased platelet reactivity, and disruption of nitric oxide metabolism. miRNA therapy commonly employs miRNA antagonists and mimics compounds to delete or restore miRNA expression, respectively. These therapies are widely used in preclinical and some clinical studies. However, the in vivo efficacy of current anti-miRNA technologies has been hindered by physiological and cellular barriers to delivering miRNA mimics/inhibitors into targeted cells and the broad regulatory nature of miRNAs, which can lead to deleterious off-target effects. Therefore, in order to overcome these challenges, further research must be committed to the method of miRNA administration to ensure precise miRNA delivery to the intended site of atherosclerosis. More studies are required in this area to explore miRNA regulatory pathways and mechanisms that might improve HHcy-atherosclerosis disease. Moreover, in order to improve the therapeutics with miRNAs, it might be interesting to develop better therapeutics together with nutritional factors such as vitamin D, folate, or curcumin, which also have a great potential to act as epigenetic modulators. Finally, miRNAs might be used as well as biomarkers in atherosclerosis disease induced by Hcy, a field that also will need further research.

## 4. Conclusions

There are several mechanisms that are well-known to be critical for HHcy-related atherosclerosis, such as endothelial dysfunction, oxidative stress, ER stress, inflammation or VSMC proliferation, and foam cell formation (see Figure 2). Interestingly, there are other types of regulations that are essential as well, which are the epigenetic ones. The most studied in HHcy-related atherosclerosis are DNA methylation and histone modifications. As far as we know, this is the first review that describes the role of different miRNAs in hyperhomocystein-atherosclerosis-related disease. However, the importance of miRNAs for HHcy-related atherosclerosis is yet in its infancy (Table 1 and Figure 2), representing a major challenge in this field. Overall, further investigation is needed to clarify the role of the different miRNAs in hyperhomocystein-mediated atherosclerosis.

## Figures and Tables

**Figure 1 ijms-23-12452-f001:**
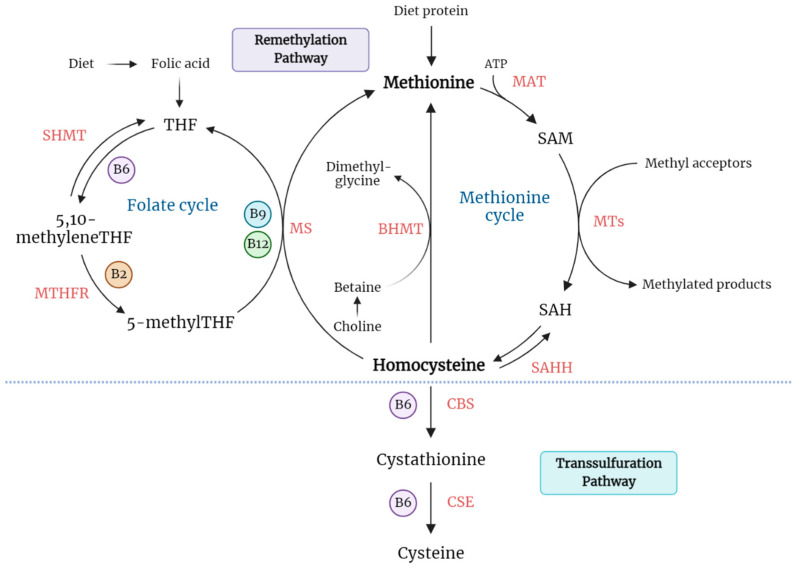
Schematic representation of homocysteine (Hcy) metabolism. Biosynthesis of Hcy occurs through the transmethylation of methionine via three sequential steps. Methionine is the only known source of Hcy and is acquired from diet protein. Metabolism of Hcy is at the intersection of two alternative pathways: it may be re-methylated back to methionine via folate-dependent/independent remethylation pathway or irreversibly degraded to cysteine via transsulfuration pathway. These pathways require vitamin-derived cofactors, including pyridoxine (vitamin B6), folate (vitamin B9), cobalamin (vitamin B12), and riboflavin (vitamin B2). Abbreviations; BHMT, betaine-homocysteine S-methyltransferase; CBS, cystathionine β-synthase; CSE, cystathionine γ-lyase; MAT, methionine adenosyltransferase; MS, methionine synthase; MT, methyltransferase; MTHFR, 5,10-methylenetetrahydrofolate reductase; SAH, S-adenosylhomocysteine; SAHH, S-adenosylhomocysteine hydrolase; SAM, S-adenosylmethionine; SHMT, serine hydroxymethyltransferase; THF, tetrahydrofolate.

**Figure 2 ijms-23-12452-f002:**
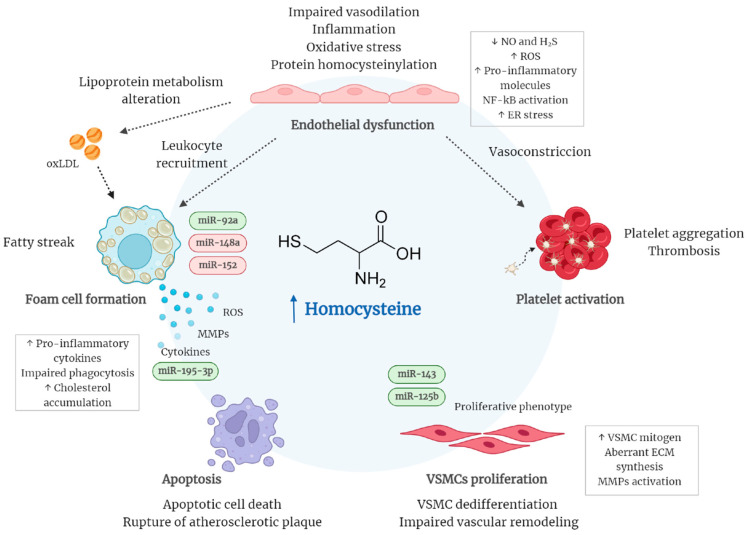
Schematic overview of the underlying mechanisms that mediate the adverse effects of HHcy in CVD. Elevated Hcy levels induce impaired vasodilation, inflammation, and oxidative stress leading to endothelial dysfunction. Oxidative stress can further stimulate platelet activation and apoptosis as well as alter lipoprotein metabolism, resulting in macrophage foam cell formation. Additionally, HHcy promotes the dedifferentiation of VSMCs into a proliferative phenotype, contributing to vascular remodeling. MicroRNAs that could be involved in some of the mechanisms have been also included. In green are those miRNAs that are downregulated and in red upregulated. Abbreviations: ECM, extracellular matrix; ER, endoplasmic reticulum; MMPs, matrix metalloproteinases; NF-κB, Nuclear factor-κB; NO, nitric oxide; oxLDL, oxidized low-density lipoprotein; ROS, reactive oxygen species; VSMCs, vascular smooth muscle cells.

**Table 1 ijms-23-12452-t001:** Effects of HHcy and miRNAs in HHcy-related atherosclerosis disease.

Studies	Treatment	Epigenetic Alterations	Effects	Cell Culture/ Animal Model
Ref. [86]	50, 100, 200, 500 µM of Hcy for 72 h	↑DNMT3a mRNA ↓miR-143	↑Proliferation of VSMCs	Human VSMCs
Ref. [87]	50, 100, 200, 500µM of Hcy for 72 h/ High-Met diet for 15 weeks	↓miR-125b ↑DNMT3b protein ↑p53 DNA methylation	↑Proliferation of VSMCs	Human VSMC/ ApoE^-/-^ mice
Ref. [88]	High-Met diet for 16 weeks	↓miR-92a ↑EZH2 mRNA ↑H3K27me3	↑Accumulation of total cholesterol and triglycerides in foam cells	ApoE^-/-^ mice
Ref. [89]	High-Met diet for 20 weeks	↓DNMT1 mRNA ↑miR-148a/152 ↑ADRP mRNA	↑Accumulation of total cholesterol and cholesteryl ester in foam cells	ApoE^-/-^ mice
Ref. [90]	High-Met diet for 16 weeks	↑DNMT3a and HDAC11 mRNA ↓miR-195-3p ↑IL-31	↑Macrophage inflammation and atherosclerotic plaque formation	ApoE^-/-^ mice

Abbreviations; ADRP, adipose differentiation-related protein; DNMT, DNA methyltransferase; EZH2, enhancer of zeste homolog 2; Hcy, homocysteine; HDAC, histone deacetylase; IL-31, interleukin-31; Met, methionine; VSMC, vascular smooth muscle cell. ↑ Increase; ↓ Decrease

## Data Availability

Not applicable.

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
