# Peer review of "Epigenetic Regulation by microRNAs in Hyperhomocysteinemia-Accelerated Atherosclerosis"

_ijms, 2022, doi:10.3390/ijms232012452_

Round 1

Reviewer 1 Report

Thank you for allowing me to evaluate this well written paper on Hcy and (epi)genitcs. 

I truly enjoyed the paper.

I have just one suggestion. The abstract ends without clear conclusions, where the text manuscript does. Please include these focussed conclusions in the abstract as well.

Very nice illustrations, otherwise I have no comments.

Author Response

Manuscript ID: ijms-1957522

Title: Epigenetic regulation by microRNAs in hyperhomocysteinemia-accelerated 
atherosclerosis

Referee 1:

I have just one suggestion. The abstract ends without clear conclusions, where the text manuscript does. Please include these focussed conclusions in the abstract as well.

We thank the Reviewer for the encouraging and positive comments. We have revised the abstract following the Reviewer’s suggestions and added a conclusion with the examples of miRNAs that have been shown to play a key role in HHcy-related atherosclerosis (lines 41-49)

Reviewer 2 Report

This study has been designed and presented already in a way that easy to follow and understandable. However, Authors need to address some minor issues:

1) Need to discuss some aspects of fatty acid and glucose metabolism to connect how aberrant metabolic system contributes to atherosclerosis

2) Which enzymes or pathways of oxidative stress generation are involved in atherosclerotic complication need to discuss. 

Author Response

Manuscript ID: ijms-1957522

Title: Epigenetic regulation by microRNAs in hyperhomocysteinemia-accelerated 
atherosclerosis

Referee 2:

1) Need to discuss some aspects of fatty acid and glucose metabolism to connect how aberrant metabolic system contributes to atherosclerosis

2) Which enzymes or pathways of oxidative stress generation are involved in atherosclerotic complication need to discuss

We thank the Reviewer’s for the careful reading of the text and constructive suggestions.

We have revised the manuscript following the Reviewer’s suggestions as detailed below

We added and discussed this in lines 208-236

We added and discussed this in lines 257-272

We added new abbreviations from these new parts